# Compact Silicon-Arrayed Waveguide Gratings with Low Nonuniformity

**DOI:** 10.3390/s24165303

**Published:** 2024-08-16

**Authors:** Chengkun Yang, Zhonghao Zhou, Xudong Gao, Zhengzhu Xu, Shoubao Han, Yuhua Chong, Rui Min, Yang Yue, Zongming Duan

**Affiliations:** 1Center for East China Research Institute of Electronics Engineering, Hefei 230036, China; gaoxudong311@126.com (X.G.); zhenzhupearl@163.com (Z.X.); shoubao_cetc@126.com (S.H.); chchyh@163.com (Y.C.); duanzongm@163.com (Z.D.); 2Center for Cognition and Neuroergonomics, State Key Laboratory of Cognitive Neuroscience and Learning, Beijing Normal University, Zhuhai 519087, China; rumi@doctor.upv.es; 3School of Information and Communications Engineering, Xi’an Jiaotong University, Xian 710049, China; yueyang@xjtu.edu.cn

**Keywords:** microwave photonics, array waveguide gratings, integrated photonic devices

## Abstract

Array waveguide gratings (AWGs) have been widely used in multi-purpose and multi-functional integrated photonic devices for Microwave photonics (MWP) systems. In this paper, we compare the effect of output waveguide configurations on the performance of AWGs. The AWG with an output waveguide converging on the grating circle had larger crosstalk and lower nonuniformity. We also fabricated a 1 × 8 AWG with an output waveguide converging onto the SOI’s grating circle, whose central operation wavelength was around 1550 nm. The fabricated AWG has a chip size of 500 μm × 450 μm. Experimental results show that the adjacent channel crosstalk is −12.68 dB. The center channel insertion loss, as well as 3 dB bandwidth, are 4.18 dB and 1.22 nm at 1550 nm, respectively. The nonuniformity is about 0.494 dB, and the free spectral range is 19.4 nm. The proposed AWG is expected to play an important role in future MWP systems given its good nonuniformity and insertion loss level.

## 1. Introduction

Microwave photonics (MWP), where photonic devices and techniques are required to generate, manipulate, transport, and measure highspeed radiofrequency (RF) signals, has attracted growing research interest [1,2,3]. Conventional MWP systems consist of relatively expensive discrete fiber optic components, which have the disadvantages of bulky size and susceptibility to external perturbations. The advent of the advanced photonic integration platform, which allows the fabrication of multi-purpose and multi-functional circuits on a photonic chip [4], offers the required framework for MWP technologies. By these means, stable, cost-effective, and scalable integrated photonic devices can be developed.

Recently, multi-purpose and multi-functional integrated photonic devices have been developed for many MWP applications, such as optical beamformers [5,6], optical true time delay [7,8], millimeter wave signal generation [9], and wavelength interrogation [10], etc. Most integrated photonic devices consist of discrete integrated photonic components, especially array waveguide gratings (AWGs). AWGs are one kind of angular dispersion passive device where multi-beam interference is exploited to introduce certain optical path differences through the array waveguide and interferometrically image different wavelengths into the slab waveguide, which is received by different output waveguides. Owing to their distinguished advantages, such as high resolution, multi-channel, high integration, low crosstalk, and low loss, AWGs have been widely applied in multi-spectral imaging [11,12], spectrum analysis [13,14,15], dense-wavelength-division multiplexing [16,17], astronomy [18], biomedical applications [19], and optical sensing [20,21].

AWGs have been developed based on various materials such as silica [22,23,24], indium phosphide (InP) [25,26], polymer [19,27], silicon nitride [28,29,30], thin-film lithium niobate (TFLN) [31,32,33], and silicon [34,35,36]. Silica-based AWGs have been successfully commercialized due to their minimal loss and mature processing technology. However, their large footprint, caused by silica’s low refractive index, hinders high-density integration. InP-based AWGs exhibit excellent optoelectronic properties and waveguide integration capabilities, allowing for efficient optical coupling and transmission. Although they benefit from relatively mature processing technology, they are hindered by high production costs, poor thermal stability, and a low damage threshold. Polymer-based AWGs offer low cost, flexibility, and ease of manufacture. However, they typically exhibit relatively low thermal stability and damage thresholds. Silicon nitride-based AWGs provide a moderately high refractive index and low absorption loss. However, they also have disadvantages, including stress issues and limited thermo-optic effects. TFLN-based AWGs use the electro-optic effect to fast-tune their operating wavelength. However, their manufacturing process is complex, and they suffer from low mechanical strength, poor thermal stability, and high costs. Low-index materials have managed to demonstrate excellent performance and commercial maturity. Furthermore, highly integrated compact silicon photonics circuits can provide high refractive index contrast in silicon nanowire waveguides, reducing the waveguide bending radius and miniaturizing the device size. Silicon photonics coms-compatible technology is also suitable for fabricating silicon-based AWGs, showing good feasibility in mass production at low cost.

AWGs can have various structural forms, with the main differences lying in the arrangement of the phase array. The common structures are S-shaped [37,38], arc-shaped [11,39,40], and horseshoe-shaped [41,42,43]. Horseshoe-shaped AWGs have a constant length for all array arms, which contributes to better performance and is straightforward from a design perspective. Arc-shaped AWGs minimize the number of waveguide junctions, but junction losses and mode conversion at these junctions can degrade performance. S-shaped AWGs exhibit lower dispersion values due to S-bend waveguide arrays, where the dispersion of one curved section is counteracted by a second section with opposite curvature [44]. Other novel structures have been proposed to improve the performance of arrayed waveguide gratings, such as overlapped AWGs [45], reflection-shaped AWGs [46], etc.

However, silicon AWGs do not perform well in terms of crosstalk and insertion loss characteristics. It should be noted that a few factors deteriorate the optical transmission characteristics of AWGs, including the coupling between single-mode waveguides and reflections induced by mode mismatching between the single-mode waveguide and the free propagation region (FPR). To solve these issues, tapers are usually used to establish efficient connections between single-mode waveguides and FPRs. The tapers in an AWG should be long enough to lower the optical loss level and widen the operation bandwidth of each channel. Different types of tapers have been developed to improve AWG performance. Linear tapers proved efficient in lowering propagation loss but ineffective in suppressing crosstalk between different channels [46,47]. Double-etched tapers can solve both propagation loss and crosstalk issues, but the fabrication procedure is complicated, where two steps of etching are normally required [34,48]. Compact parabolic tapers fabricated based on simple procedures have been employed in AWGs to reduce insertion loss and suppress crosstalk at the user terminal [43,49]. The loss of uniformity directly affects its high-speed modulation performance, tolerable wavelength drift, and the performance of cascaded systems. Several methods have been investigated to improve the loss uniformity, such as 2 × 1 multimode interference couplers [50], mode-field converters [51], auxiliary waveguides [52], etc.

In this study, we designed and fabricated a 1 × 8 arc-shaped AWG on a SOI platform with a central operation wavelength around 1550 nm. Parabolic tapers were introduced to bridge the single-mode waveguides and FPRs for better insertion loss and crosstalk performances while maintaining chip compactness. We compared the effect of output waveguide configurations on the performance of AWGs. Then, we numerically simulated the impacts of the array waveguide number on its optical transmission characteristics to optimize the primary parameters of AWGs. We fabricated AWGs based on the simulation results, and experimental results indicated that the AWGs outperform in insertion loss and nonuniformity.

## 2. Theory and Design

An AWG consists of input/output waveguides, input/output star coupler, and array waveguides. Array waveguides connect input/output star couplers with identical Rowland circle designs through a constant path length difference between neighboring waveguides. AWG principles are described as follows: When wideband light is launched into the input star coupler from the central waveguide, it diffracts and forms a Gaussian optical field. This field diverges into each channel of array waveguides. Then, light propagates in the arrayed waveguides and arrives at an output aperture with a phase difference of $n_{\mathrm{eff}}k_{0}∆L$ between the neighboring waveguides, where $k_{0}=2\pi /\lambda$ is the wavenumber and n_eff_ is the effective refractive index of the fundamental mode propagating in each waveguide. After transmitting through array waveguides, the light output diverges in the output coupler and shifts along the image plane, with the wavelength deviating from the central wavelength *λ*_c_. The output waveguides are arranged over this image plane and each collects one section of the spatially separated spectrum. The input and output waveguides satisfy the equation:(1)$$n_{s}d_{a}\sin\theta_{i}+n_{\mathrm{eff}}∆L+n_{s}d_{a}\sin\theta_{o}=m\lambda$$

For central input, $\theta_{i}$ and $\theta_{o}$ tend to 0, the constant path length difference between adjacent array waveguides satisfies the equation:(2)$$∆L=m\lambda_{c}/n_{\mathrm{eff}}$$

The pitch of output waveguides is provided by the equation:(3)$$d_{o}=\frac{∆\lambda \mathrm{Dm}}{d_{a}}\frac{n_{g}}{n_{\mathrm{eff}}n_{s}}$$
where $d_{a}$ is the distance between the adjacent array waveguides; Δλ is the channel spacing; D is the diameter of the Rowland circle; m is the diffraction order; $n_{g}$ is the group index of the fundamental mode of the arrayed waveguide; $n_{s}$ is the effective index of the FPR; $\theta_{i}$ and $\theta_{o}$ are the angles between the input/output waveguide and the central channel waveguide, respectively.

The structural design of the taper between the single-mode waveguide and FPR is an important issue affecting AWG performance. Compact parabolic tapers reduce insertion loss and receiver crosstalk in AWGs. In this study, the parabolic taper was optimized to improve AWG performance. The profile of the parabolic taper was designed using the following equation [11]:(4)$$y=A({\mathrm{Lt}-x)}^{m}+B$$
where A and B are constant, as determined by the waveguide width and taper width; m is the power of the exponent and Lt is the total taper length.

Since the waveguide and receiver aperture designs already determine the input and output waveguide widths, only two parameters need optimization: the length of the taper and its geometry, which is defined by the exponent m. In the study [11], various taper geometries with m ranging from 1 to 4 were simulated. The authors found that the parabolic taper with m *=* 2 achieved the most efficient adiabatic coupling, as shown in Figure 1a. Figure 1c shows the transmission of the parabolic taper was investigated for different taper lengths. In these simulations, light was defined to propagate from the wide to the narrow waveguide. The mode conversion efficiency increased with increments in taper length and a 5.0 μm long parabolic taper achieved transmission over 97%. The simulated field intensity for a parabolic taper length of 5 μm at a wavelength of 1550 nm is shown in Figure 1b.

To investigate the effect of output waveguide configurations on the performance of AWGs, two AWG types were designed, as shown in Figure 2. Both include Rowland circle designs and have the same parameters except for the output waveguide configurations. In the Type-1 design, the input/output waveguides all point toward the center of the Rowland circle with a radius of D/2, as shown in Figure 2a. In the Type-2 design, the input/output waveguides lie on the Rowland circle and converge on the grating circle, as shown in Figure 2b. The array waveguides of two AWG types lie on the grating circle with a radius of D.

Channel spacing and the number of output waveguides are specified by Δ*λ* and *N_o_*, respectively. The widths of the input waveguide, arrayed waveguides, and output waveguides are set to the same value. The distance between the adjacent array waveguides (*d_a_*) is determined by the fabrication resolution of the lithography. Other parameters of the AWG are calculated based on the theoretical model in [28]. The optimal parameters of the AWG are shown in Table 1. The AWG transmission spectrum is simulated based on the 2.5-dimensional finite–difference time–domain (FDTD) method, showing far-field projection across the star coupler and phase errors through the array waveguide.

The number of array waveguides impacts insertion loss, adjacent channel crosstalk, nonuniformity, and overall device size. Adjacent channel crosstalk is defined as the difference between the peak insertion loss within the channel passband and the minimum insertion loss within the passband of adjacent channels. Conversely, non-adjacent channel crosstalk is the difference between the peak insertion loss within the channel passband and the minimum insertion loss within the passband of all non-adjacent channels. According to the International Telecommunication Union standard ITU-T G.671, the channel passband width is defined as a quarter of the AWG channel spacing. As shown in Figure 3, with increasing array waveguides, the insertion loss of AWGs decreases and the adjacent channel crosstalk of the AWG increases. The reason for this is that the light field at the receiving end intensifies with increasing arrayed waveguide increments, thereby lowering AWG insertion loss to some extent. However, with the increased number of array waveguides, the phase error introduced through the fabrication process may also increase the adjacent channel crosstalk of the device. Upon comparing AWG to two output waveguide configurations, output waveguide configurations affect crosstalk and nonuniformity. Type-2 has a lower nonuniformity. Both configurations do not affect the insertion loss. Simulated spectral responses for the AWG with 20 array waveguides are shown in Figure 4. From Figure 4a,b, it can be concluded that the adjacent channel crosstalk is −14.69 dB. The insertion loss and 3 dB bandwidth of the central channel are 3.25 dB and 1.09 nm at 1550 nm, respectively. The transmission nonuniformity is about 1.96 dB, and the free spectral range is 19.42 nm. From Figure 4c,d, it can be concluded that the adjacent channel crosstalk is −15.28 dB. The insertion loss and 3 dB bandwidth of the central channel are 3.17 dB and 1.08 nm at 1550 nm, respectively. The transmission nonuniformity is about 0.627 dB, and the free spectral range is 19.42 nm.

## 3. Measurements and Discussion

For comparison, the AWG with output waveguides converging on the grating circle (Type-2) was chosen for fabrication. The optimally designed AWGs were fabricated by Chongqing United Microelectronics Center (CUMEC, Chongqing, China) using the CSiP180AL technology platform. This platform provides MPW services based on standard SOI with 2 μm BOX and 220 nm top silicon. Its silicon photonic chip manufacturing processes are compatible with standard CMOS processes, including deep UV lithography and inductively coupled plasma dry etching. The microscopic image of the AWG and enlarged microscope images of the output star coupler and parabolic tapers are shown in Figure 5. The total footprint of the fabricated AWG is 500 μm × 450 μm. A coupling platform was used to test the performance of the fabricated device. A broadband light from a superluminescent diode (SLD) was launched into a single-mode fiber and then coupled to the input waveguide via a grating coupler. The light from the output waveguide was connected to an optical spectrum analyzer (OSA, AQ6370C, Yokogawa, Japan) with single-mode fibers to collect transmission light.

The transmission spectrum of the AWG is obtained by subtracting measurements of the same grating coupler taken back-to-back, as shown in Figure 6. As shown in Figure 6b, the adjacent channel crosstalk is −12.68 dB and the non-adjacent channel crosstalk is −14.66 dB. The center channel insertion loss and 3 dB bandwidth are 4 dB and 1.22 nm for the central output port, respectively. For the edge output port, the adjacent channel crosstalk is −12.15 dB and the non-adjacent channel crosstalk is −15.32 dB, as shown in Figure 6c. The nonuniformity is about 0.494 dB and the free spectral range is 19.4 nm.

The nonuniformity matches well with the simulation results while the channel central wavelength shifts by 0.79 nm, the insertion loss deteriorates by 1.01 dB and the channel crosstalk varies by 2.6 dB, are shown in Table 2. The AWG’s crosstalk and insertion loss is worse than mainstream AWGs. The bending loss of the arrayed waveguides is not considered in the simulation process because the array waveguides normally experience large bending. Fabrication error is the main reason accounting for the slight deviation in measurement results from the originally designed values. Due to fabrication errors, the sidewall of the waveguides is no longer smooth, resulting in large scattering losses. The roughness of the waveguides is the largest factor that deteriorates the insertion loss feature. Additionally, the crosstalk is more sensitive to the sidewall roughness. The incremental overlap between the waveguide guiding mode and the sidewall of the waveguide leads to higher crosstalk. In the designed AWG, the width of the bent waveguide is narrower than the waveguide width of mainstream AWGs. As the waveguide becomes narrower, the guiding mode in the bent waveguide tends to strongly overlap with the sidewall of the waveguides, causing larger phase errors and higher crosstalk. Due to fabrication errors, the refractive index of the waveguide may deviate from the design parameters, causing the experimentally central wavelength to shift from the original designed value.

## 4. Conclusions

In conclusion, a 1 × 8 AWG on a SOI platform with a 1550 nm operation wavelength was designed and fabricated. The output waveguide configurations of AWGs affect crosstalk and nonuniformity. The AWG with an output waveguide converging on the grating circle had larger crosstalk and lower nonuniformity. The fabricated AWG with an output waveguide converged on the grating circle had an area of 500 μm × 450 μm. Experimental results indicated that the single-channel crosstalk was −12.68 dB, and the center channel insertion loss and 3 dB bandwidth were 4.18 dB and 1.22 nm at 1550 nm, respectively. The nonuniformity was about 0.494 dB, and the free spectral range was 19.4 nm. For MWP systems, such as optical beamformers, optical true time delay, millimeter wave signal generation, and wavelength interrogation, silicon AWGs were expected to have the lowest crosstalk and insertion loss. The specific requirements for crosstalk and insertion loss performance in silicon AWGs depend on the system parameters, which are still under investigation and will be reported in the near future.

## Figures and Tables

**Figure 1 sensors-24-05303-f001:**
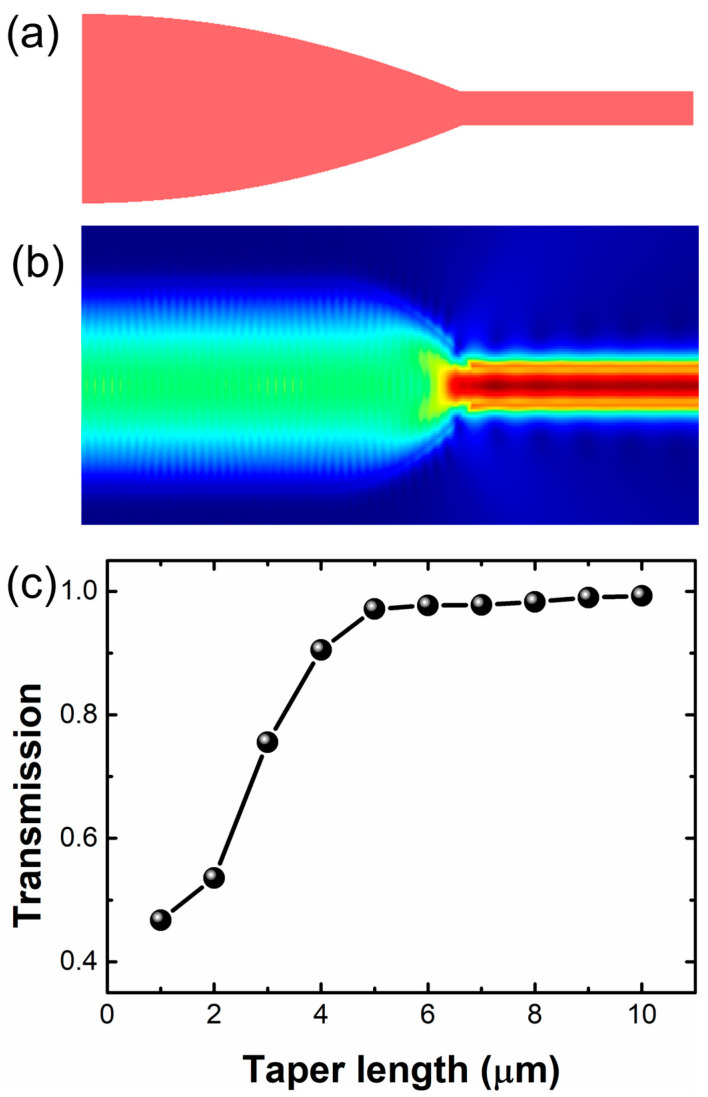
(**a**) Schematic diagram of a parabolic taper; (**b**) simulated field intensity for a taper length of 5 μm at a wavelength of 1550 nm; (**c**) transmission of tapers as a function of taper length at a wavelength of 1550 nm.

**Figure 2 sensors-24-05303-f002:**
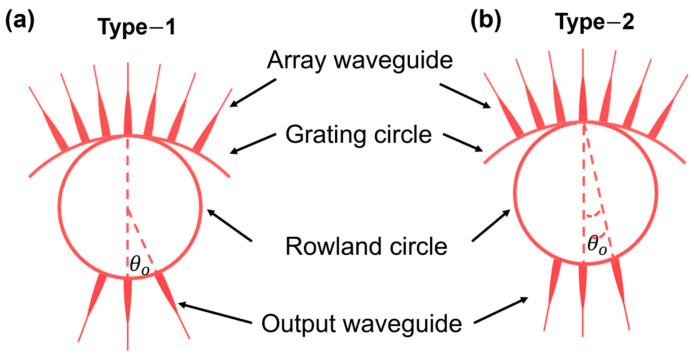
Schematic layout of output star coupler of the AWG structure. (**a**) Type-1 output waveguides all point toward the center of the Rowland circle; (**b**) Type-2: output waveguide converges on the grating circle.

**Figure 3 sensors-24-05303-f003:**
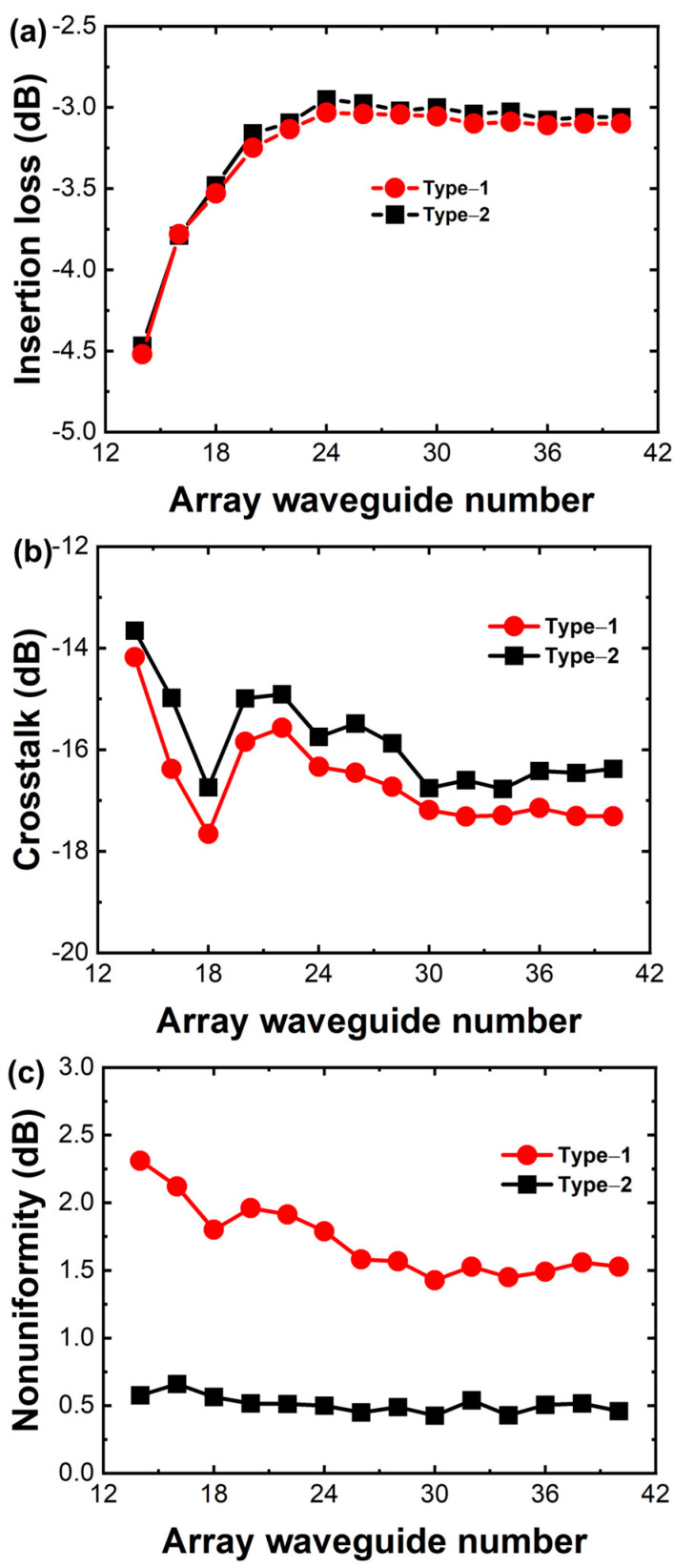
Simulated AWG performance as a function of array waveguides. (**a**) Insertion loss; (**b**) adjacent channel crosstalk; (**c**) nonuniformity.

**Figure 4 sensors-24-05303-f004:**
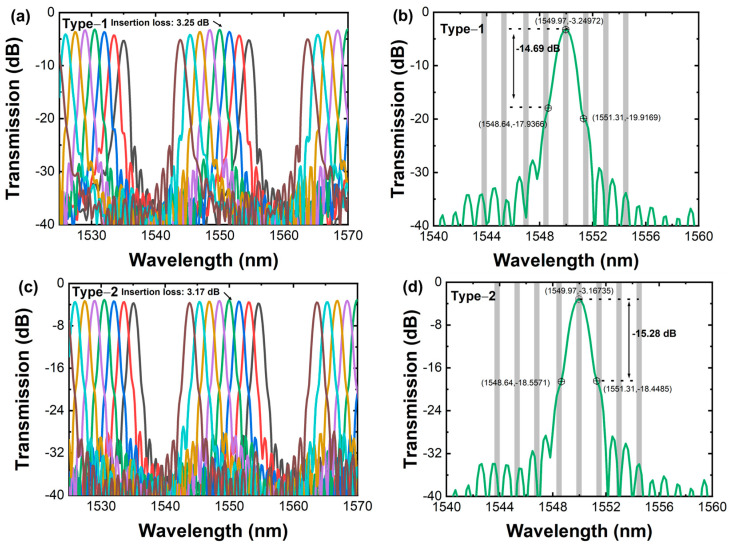
Type-1: (**a**) simulated spectral responses for the AWG with 20 array waveguides; (**b**) simulated spectral responses for the central output port. Type-2: (**c**) simulated spectral responses for the AWG with 20 array waveguides; (**d**) simulated spectral responses for the central output port.

**Figure 5 sensors-24-05303-f005:**
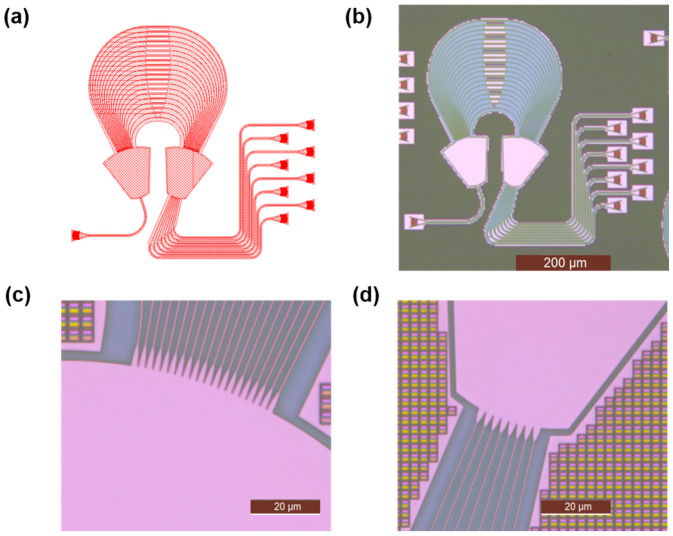
Illustrations of the fabricated AWGs. (**a**) A snippet from the GDSII mask layout file of an AWG; (**b**) microscopic picture of an AWG; (**c**) microscopic picture of the parabolic tapers between the FPR and array waveguides; (**d**) microscopic picture of the parabolic tapers between the output waveguides and FPR.

**Figure 6 sensors-24-05303-f006:**
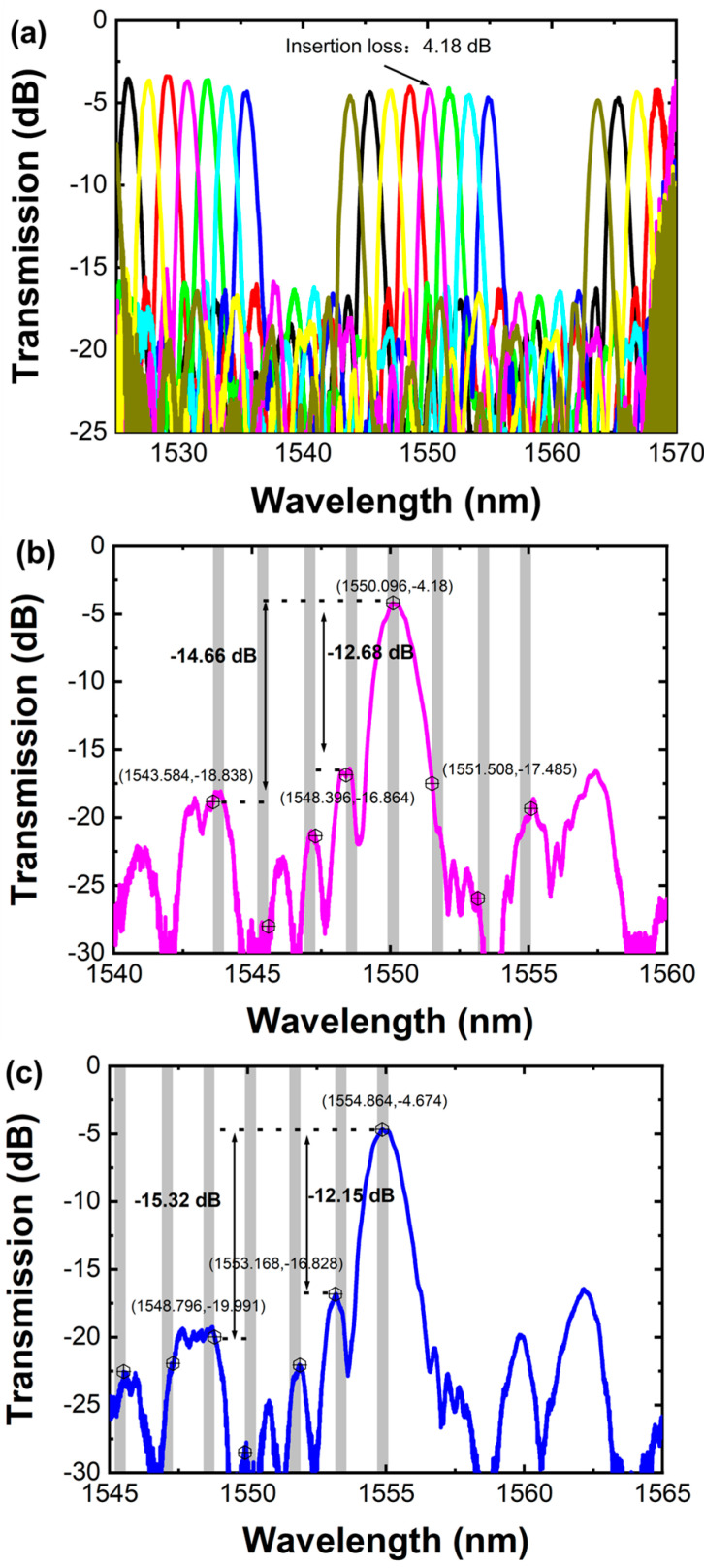
(**a**) Transmission spectrum for all output ports; (**b**) the central output port; (**c**) the edge output port.

**Table 1 sensors-24-05303-t001:** Primary design parameters of the AWG.

Parameter	Symbol	Value	Units
Channel spacing	Δλ	1.6	nm
Central wavelength	λ_c_	1549.3	nm
Length increment	ΔL	28.29	μm
Number of input/output channels	N_i_/N_o_	1/8	
Spacing of input/output waveguide	d_i_/d_o_	2.7	μm
Spacing of array waveguides	d_a_	1.7	μm
Diffraction order	m	43	
Gap between adjacent tapers	g_a_	0.2	μm
Length of the tapers	L_t_	5	μm
Diameter of Rowland circle	D	103	μm
Waveguide width	w	0.45	μm

**Table 2 sensors-24-05303-t002:** Simulation and measurement results of the AWG.

Parameter	Simulation	Measurement	Units
Central wavelength	1549.3	1550.09	nm
Insertion loss	3.17	4.18	dB
Crosstalk	−15.28	−12.68	dB
Nonuniformity	0.627	0.494	dB
3-dB bandwidth	1.08	1.22	dB
FSR	19.42	19.4	nm

## Data Availability

Data underlying the results presented in this paper are not publicly available at this time but may be obtained from the authors upon reasonable request.

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
