# Peer review of "Compact Silicon-Arrayed Waveguide Gratings with Low Nonuniformity"

_sensors, 2024, doi:10.3390/s24165303_

Round 1

Reviewer 1 Report

Comments and Suggestions for Authors

The authors presented a 1×8 Arc-shaped AWG on SOI platform with central operation wavelength around 1550 nm. In the manuscript,the authors simulated the impacts of the array waveguide number on its optical transmission characteristics, and the primary parameters of AWG are optimized. The experimental performance show that the adjacent channel crosstalk is -12.68 dB, and the center channel insertion loss, as well as 3 dB bandwidth are 4.18 dB and 1.22 nm at 1550 nm, respectively, which is basically in agreement with the simulation.

The reviewer believes in the novelty and quality of the work and recommends publishing in the Sensers edition provided that the authors consider the following comments / questions:

1. The authors claim that the insertion loss of their experimental test is 4.18 dB, which I am skeptical. Since the state-of-the-art grating coupling structure shows that the loss from input to output is much greater than this value, the author should explain the calculation and normalization method of the insertion loss in the manuscript.

2. Pay attention to the format of the letters in the formula, many of the letters in the formula on the third page of the author are mixed with regular and italics, such as m,ΔL. Please proofread them in detail.

3. Please carefully proofread the manuscript to avoid some simple mistakes, such as two A and two B in Figure 4.

4. The authors use the grating coupling method to test the transmission spectrum of AWG, but in principle, one of the defects of grating coupling is that the wavelength is more sensitive, how to input a wide spectrum light source at the same time, to ensure that the output spectral intensity is consistent. I suggest that this be explained in the testing section.

5. The authors should further identify more specific potential applications.

Author Response

Comment1:The authors claim that the insertion loss of their experimental test is 4.18 dB, which I am skeptical. Since the state-of-the-art grating coupling structure shows that the loss from input to output is much greater than this value, the author should explain the calculation and normalization method of the insertion loss in the manuscript.

Reply1:Thanks for your constructive suggestion.  The following descriptions have been added in the revised manuscript:

“The transmission spectrum of the AWG is obtained by subtracting measurements of the same grating coupler taken back-to-back, as shown in Figure 6.”

Comment2:Pay attention to the format of the letters in the formula, many of the letters in the formula on the third page of the author are mixed with regular and italics, such as m,ΔL. Please proofread them in detail.

Reply2:Thanks for your carefully review. We have been carefully checked the formula in the revised manuscript.

Comment3:Please carefully proofread the manuscript to avoid some simple mistakes, such as two A and two B in Figure 4.

Reply3:Thanks for your carefully review. We have been carefully checked mistakes in the revised manuscript.

"Figure 4. Type-1: (a) simulated spectral responses for the AWG with 20 array waveguides, (b) simulated spectral responses for the central output port; Type-2: (c) simulated spectral responses for the AWG with 20 array waveguides, (d) simulated spectral responses for the central output port."

Comment4:The authors use the grating coupling method to test the transmission spectrum of AWG, but in principle, one of the defects of grating coupling is that the wavelength is more sensitive, how to input a wide spectrum light source at the same time, to ensure that the output spectral intensity is consistent. I suggest that this be explained in the testing section.

Reply4:Thanks for your constructive suggestion. Method to reduce the impact of grating couplers in chip testing : reference optical path: set up a reference optical path during testing to measure the inherent loss of the grating coupler, and then perform calibration and compensation in data analysis; power normalization: normalize the measured optical power to eliminate the impact of coupling loss.

Comment5:The authors should further identify more specific potential applications.

Reply5:Thanks for your constructive comments. The following descriptions have been added in the revised manuscript:

For MWP systems, such as optical beamformer, optical true time delay, millimeter wave signal generation, and wavelength interrogation, the crosstalk and insertion loss of silicon AWGs are expected to be the lower the better. The specific requirements for crosstalk and insertion loss performance in silicon AWGs depend on the system parameters, which are still under investigation and will be reported in the near future.

Reviewer 2 Report

Comments and Suggestions for Authors

In this research paper, authors addressed the issues related to crosstalk and insertion loss of silicon photonics by designing a 1×8 Arrayed Waveguide Grating (AWG) on a Silicon-On-Insulator (SOI) platform, operating at a wavelength of 1550 nm. At the beginning, author briefly introduced the status quo of the field. They discussed various materials and structural designs for AWGs, including silica, indium phosphide (InP), polymer, silicon nitride, thin-film lithium niobate (TFLN), and silicon, for their integration capabilities, cost-effectiveness, thermal stability, and mechanical properties.  Through simulation and experimental validations, authors demonstrate great results, including  a crosstalk of -12.68 dB for adjacent channels, an insertion loss of 4 dB, etc.. However, there are 2-3 points are still not clear enough to me. I suggest editor to accept the manuscript after a minor revision. Authors can also leverage this opportunity to further improve their draft.

Major:

1.      P8 L245, Since the fab technology accounts for the biggest difference between simulation and experimental results, could you add some detail on the fab part? Like what litho, etch process were used; How the fab error/sidewall roughness approximately is?

2.      P8 L241-255 and P9 L261-275 are exactly seem. But the value comparison between simulation and experimental results are significantly different. I suggest authors carefully check the data both in their experimental record and the table in manuscript. Make sure to provide readers with the accurate results.

3.      P10 L285-286. Authors mentioned that “AGWs usually do not perform well in terms of crosstalk and insertion loss characteristics” in P2 L86-87. How do author evaluate their design compared to the mainstream AGWs? I suggest author add some comment on if their new design is better or worse than the mainstream AGWs regarding the crosstalk/insertion loss.

Minor:

1.      P2 L73, should it be CMOS-compatible?

2.      P7 L219, what does CUMEC stand for?

Author Response

Major

Comment1:P8 L245, Since the fab technology accounts for the biggest difference between simulation and experimental results, could you add some detail on the fab part? Like what litho, etch process were used; How the fab error/sidewall roughness approximately is?

Reply1:Thanks for your constructive comments. The following descriptions have been added in the revised manuscript:

“The optimally designed AWGs are fabricated by Chongqing United Microelectronics Center (CUMEC, China) on the CSiP180AL technology platform. This platform provides MPW services based on standard SOI with 2μm BOX and 220 nm top silicon. Its silicon photonics chip manufacturing processes are compatible with standard CMOS processes, including deep UV lithography and inductively coupled plasma dry etching.”

Comment2:P8 L241-255 and P9 L261-275 are exactly seem. But the value comparison between simulation and experimental results are significantly different. I suggest authors carefully check the data both in their experimental record and the table in manuscript. Make sure to provide readers with the accurate results.

Reply2:Thanks for your carefully review. We have been removed similar content and carefully checked values in the revised manuscript.

Comment3: P10 L285-286. Authors mentioned that “AGWs usually do not perform well in terms of crosstalk and insertion loss characteristics” in P2 L86-87. How do author evaluate their design compared to the mainstream AGWs? I suggest author add some comment on if their new design is better or worse than the mainstream AGWs regarding the crosstalk/insertion loss.

Reply3: Thanks for your constructive suggestion.  The crosstalk and insertion loss of AWG is worse than the maintream AGWs. In the article, we analyzed the reasons:

“Compared with the simulation results, the nonuniformity well matches while chan-nel central wavelength shifts by 0.79 nm, the insertion loss deteriorates by 1.01 dB and the channel crosstalk shows 2.6 dB variation. Crosstalk and insertion loss of AWG is worse than the maintream AGWs. The bending loss of the arrayed waveguides is not considered in the simulation process because the array waveguides normally experience large bending. Fabrication error is the main reason that accounts for the slight deviation of the measurement results from the originally designed values. Due to fabrication error, the sidewall of the waveguides becomes no longer smooth, resulting in large scattering losses. Roughness of the waveguides is the largest factor that deteriorates the insertion loss feature. Besides, the crosstalk is more sensitive to the sidewall roughness. The increment of the overlap between the waveguide guiding mode and the sidewall of the waveguide would lead to higher crosstalk. In the designed AWG, width of the bend waveguide is narrower the waveguide width of the mainstream AGWs. As the waveguide becomes narrower, the guiding mode in the bend waveguide tend to strongly overlap with the sidewall of the waveguides, and thus to cause larger phase errors and higher crosstalk. Due to the fabrication error, the refractive index of the waveguide may deviate from the design parameters, causing the experimentally central wavelength to shift from the original designed value.”

Minor

Comment1: P2 L73, should it be CMOS-compatible?

Thanks for your constructive suggestion. The descriptions have been added in the revised manuscript:

“The optimally designed AWGs are fabricated by Chongqing United Microelectronics Center (CUMEC, China) on the CSiP180AL technology platform. This platform provides MPW services based on standard SOI with 2μm BOX and 220 nm top silicon. Its silicon photonics chip manufacturing processes are compatible with standard CMOS processes, including deep UV lithography and inductively coupled plasma dry etching.”

Comment2: P7 L219, what does CUMEC stand for?

Reply2: Thanks for your constructive suggestion. The descriptions have been added in the revised manuscript:

“The optimally designed AWGs are fabricated by Chongqing United Microelectronics Center (CUMEC, China) on the CSiP180AL technology platform. This platform provides MPW services based on standard SOI with 2μm BOX and 220 nm top silicon. Its silicon photonics chip manufacturing processes are compatible with standard CMOS processes, including deep UV lithography and inductively coupled plasma dry etching.”

Round 2

Reviewer 1 Report

Comments and Suggestions for Authors

I have no more comments and suggestions

Reviewer 2 Report

Comments and Suggestions for Authors

Authors' response and revision addressed my previous comments and concerns. I recommand it to be published in Sensors.